# Dual Role of ACBD6 in the Acylation Remodeling of Lipids and Proteins

**DOI:** 10.3390/biom12121726

**Published:** 2022-11-22

**Authors:** Eric Soupene, Frans A. Kuypers

**Affiliations:** Department of Pediatrics, Division of Hematology, Children’s Hospital Oakland Research Institute, University of California at San Francisco, Oakland, CA 94609, USA

**Keywords:** myristoylation, palmitoylation, acylation, lysophospholipids, lipid droplets

## Abstract

The transfer of acyl chains to proteins and lipids from acyl-CoA donor molecules is achieved by the actions of diverse enzymes and proteins, including the acyl-CoA binding domain-containing protein ACBD6. N-myristoyl-transferase (NMT) enzymes catalyze the covalent attachment of a 14-carbon acyl chain from the relatively rare myristoyl-CoA to the N-terminal glycine residue of myr-proteins. The interaction of the ankyrin-repeat domain of ACBD6 with NMT produces an active enzymatic complex for the use of myristoyl-CoA protected from competitive inhibition by acyl donor competitors. The absence of the ACBD6/NMT complex in ACBD6.KO cells increased the sensitivity of the cells to competitors and significantly reduced myristoylation of proteins. Protein palmitoylation was not altered in those cells. The specific defect in myristoyl-transferase activity of the ACBD6.KO cells provided further evidence of the essential functional role of the interaction of ACBD6 with the NMT enzymes. Acyl-CoAs bound to the acyl-CoA binding domain of ACBD6 are acyl donors for the lysophospholipid acyl-transferase enzymes (LPLAT), which acylate single acyl-chain lipids, such as the bioactive molecules LPA and LPC. Whereas the formation of acyl-CoAs was not altered in ACBD6.KO cells, lipid acylation processes were significantly reduced. The defect in PC formation from LPC by the LPCAT enzymes resulted in reduced lipid droplets content. The diversity of the processes affected by ACBD6 highlight its dual function as a carrier and a regulator of acyl-CoA dependent reactions. The unique role of ACBD6 represents an essential common feature of (acyl-CoA)-dependent modification pathways controlling the lipid and protein composition of human cell membranes.

## 1. Introduction

The acyl-CoA binding protein ACBD6 interacts with the N-Myristoyltransferase (NMT) enzymes to form a dimeric enzymatic complex regulating and controlling the specificity of the myristoylation process [1,2,3]. N-myristoylation is an essential modification regulating the functions, stability, and membrane association of a diverse set of cytosolic proteins in cells [4,5,6,7,8,9,10,11,12]. Acylation of N-terminal glycine with the 14-carbon acyl donor myristoyl-CoA (Myr-CoA) occurs mainly during translation. Interaction with NMT requires the C-terminal ankyrin-repeat domain of ACBD6 to form an enzymatic complex with enhanced activity and protected from competitive inhibition by more abundant acyl-CoAs, such as palmitoyl-CoA [2,13,14,15,16,17,18]. Myristate and myristate analogs must be esterified with CoA to access the acyl-CoA binding site of NMT [17,19,20,21]. Upon thio-esterification by cellular acyl-CoA synthetases (ACSL), Myr analogs can compete with Myr-CoA binding and occupy the site until the analog chain can be transferred onto a polypeptide acyl-acceptor. The acyl chain of palmitoyl-CoA, which binds to NMT with high affinity, is not a substrate of the acyl-transferase reaction and the NMT catalytic cycle is blocked [1,17,19]. Similarly, the 2-hydroxymyristate chain of the 2-OH Myr-CoA analog efficiently inhibits N-myristoylation in various cell types with an estimated in vitro Ki of 45 nM, which is ≈45,000 lower than the Ki of the unesterified form [3,22,23,24,25,26,27,28,29,30,31,32], bringing into question the rationale for suggesting that the fatty acid 2-OH Myr itself could have been an inhibitor of NMT enzymes [33]. The commonly used myristoyl in vivo labeling probes 12-azidododecanoic acid (12-ADA) and 13-tetradecynoic acid (YnMyr) are also Myr analogs once converted to CoA ester derivatives by the cellular ACSL enzymes [5,7,34,35,36]. These probes designed to monitor the myristoylation of proteins actually compete with the binding of the correct acyl-donor (Myr-CoA) to NMT, and the consequence of such competition on the membrane association and functions of the thousands of myr-proteins is often overlooked during prolonged in vivo labeling experiments [37,38,39].

We established that one of the functions of ACBD6 was to protect the NMT/ACBD6 complex from competition and provide enhanced activity under Myr-CoA limiting concentrations. It was argued that binding of Myr-CoA to ACBD6 in a complex with NMT could sequester the acyl-donor away from NMT, allowing access of other acyl-CoAs to the Myr-CoA binding site and promote lack of specificity of the myristoylation reaction [40]. However, the Myr-CoA bound to the ACB domain is channeled to NMT and the acyl-CoA binding (ACB) domain is not necessary to provide protection [2,3]. The ACB domain appears to positively regulate the function of the ANK module in the ACBD6/NMT complex. Ligands bound to the ACB domain act as positive effectors of the acyltransferase activity of the NMT/ACBD6 complex. Unique among the members of the ACBD family [41], the phosphorylation of two serine residues of the ACB domain regulates the binding activity of ACBD6 and further enhances the activity of the ACBD6/NMT complex [2].

In addition to its role in regulating the function of the NMT enzymes of human and other organisms, ACBD6 can regulate the availability of acyl-CoAs in partition between the cytosolic and membrane compartments of the cells. Acyl-CoAs bound to ACBD6 are acyl-donors for the lysophospholipid acyltransferase enzymes (LPLAT). The dynamic binding property of the ACB domain allows the controlled release of acyl-CoA to the membrane-bound enzymes and protects them from the detergent-like property of their substrates [42,43].

The importance of ACBD6 is underscored by the fact that genetic mutations of the ACBD6 gene preventing the production of a full-length protein are associated with neurodegenerative syndromes in humans [3]. In view of the variety of processes influenced by ACBD6, and the presence of several other acyl-CoA binding proteins with potentially overlapping function, as well as the independent role of the two functional modules (ACB and ANK), we investigated the effects of the disruption of the ACBD6 gene in human cells. HeLa ACBD6.KO cells that do not produce the ACBD6 protein were viable. The acylation of lipids was significantly reduced in the absence of ACBD6 and the defect in PC formation by the Lands’ cycle led to a defect in formation of lipid droplets. The deficiency of the myristoylation reaction in the absence of an NMT/ACBD6 complex was evidenced by a decrease of the in vivo rate of protein myristoylation, and by the increased sensitivity of the cells to NMT inhibitors. These results established that ACBD6 supports two distinct acyl-CoA dependent acylation pathways essential for the remodeling of lipids in membranes by the LPLAT enzymes and of proteins by the NMT enzymes.

## 2. Materials and Methods

### 2.1. Construction and Characterization of the ACBD6.KO Cells

Deletion of the human ACBD6 gene was performed in HeLa cells with an ACBD6 CRISPR/Cas9 set (Santa Cruz; #sc-413630) designed to remove the entire ACB domain and generate a codon frameshift leading to disruption of the open reading frame. Transfected cells were selected in the presence of 1 µg/mL puromycin with medium changed every two days for a period of two weeks. Several surviving clones were obtained, and were initially analyzed by RT-PCR to confirm disruption. The absence of production of an ACBD6 protein, full-length or truncated, was further confirmed by Western blotting. The ACBD6.KO cells do not display apparent growth defects and were maintained in culture as their parent cells. Total RNA was isolated with PureLink RNA Mini Kit according to the manufacturer instructions (Thermo Fisher Scientific, Pittsburgh, PA, USA). Purified RNAs were treated with RNase-free DNase I (TURBO DNase, Thermo Fisher Scientific). Synthesis of cDNA was performed with the RevertAid First Strand cDNA Synthesis kit in the presence of oligo(dT) primers (Thermo Fisher Scientific). End-point RT-PCRs were performed with SuperScript™ One-Step RT-PCR System with Platinum™ Taq DNA Polymerase (Thermo Fisher Scientific). cDNA of five clones were sequenced to confirm the identity of the deletion in the clones. Western blot detection of ACBD6 (#MA5-28990; Thermo Fisher Scientific), NMT2 (#sc-136005; Santa Cruz Biotech, Dallas, TX, USA) and ACTB (#sc-81178; Santa Cruz Biotech) were performed with mouse monoclonal antibodies.

### 2.2. Cell Culture and Growth Experiments

Cells were grown in high-glucose DMEM supplemented with 10% fetal bovine serum, 2 mM glutamine, 5 mM non-essential amino acids and 1% (*v*/*v*) MEM vitamins (Thermo Fisher Scientific). Growth measurements were performed in 96-well plates and quantified by staining with the sulforhodamine B (SRB) dye, as previously described [3]. Absorbance was measured at 560 nm with a microplate reader. As indicated in the legend of figures, cells were grown in the presence of increasing concentrations of 2-hydroxymyristate (Cayman, Ann Arbor, MI, USA) and IMP-1088 (Cayman).

### 2.3. Lipid Droplet Quantification and Isolation

Lipid droplets (LDs) detection and quantification were performed with the Cell Navigator Fluorimetric Lipid Droplet assay kit (em/ex 550/640; ATT Bioquest, Pleasanton, CA, USA), according to the manufacturer’s instructions. Cells were grown in 96-well plates for 24 h to near confluency in the absence or presence of 200 µM oleic acid, made from a solution of 200 mM oleic acid/40 mM defatted BSA in PBS. LDs were quantified from set of 7 wells of three independent experiments. Cells were then fixed with TCA and stained with SBR for total protein quantification, which was used to normalize the fluorescence value of each well. LDs were isolated from cells grown to near confluency in four T75 flasks in the presence of 100 µM oleic acid for 24 h, using the LD Isolation kit (Cell Biolabs, San Diego, CA, USA). LDs were collected in about 400 µL at a protein concentration of 0.1 µg/µL and stored at −80 °C.

### 2.4. N-myristoyltransferase Activity Measurements

The reactions were performed as previously described with few modifications [2]. For the measurements performed in the presence of purified human ACSL6 (150 nM) [44], ATP (10µM) and CoASH (0.3µM) were added to the reactions. Some reactions were performed in the presence of the fatty acid precursor C_14:0_ (Myr; 20 µM), the fatty acid analog competitor 2-hydroxymiristate (2-OH Myr; 10, 50, 100, 1000 µM), the peptide binding inhibitor IMP-1088 (10, 100, 1000 nM), as indicated in the legend of the figures. Fatty acids, dried from 100 mM stock solutions made in ethanol, were maintained in solution with Triton X-100 (final concentration in reaction was 0.04%). Reactions were performed in triplicate in 200 µL at 37 °C with 250 nM purified human NMT2, unless otherwise indicated. Detection and quantification of the formation of the acyl-peptide was performed by reverse phase HPLC [2]. The measurement of the in vivo myristoyltransferase activity of the ACBD6.KO cells was performed by quantification of the incorporation of the analogue 12-azidododecanoic acid (12-ADA) (Click-iT myristic acid kit #C10268; Thermo Fisher Scientific) into proteins. After protein extractions, the azido-myristoylated proteins were reacted with alkyne-biotin (Click-iT Biotin protein analysis detection kit #C33372; Thermo Fisher Scientific) and detected by Western blotting with HRP-Streptavidin. Cells were grown in T75 flasks to about 70% confluency, and 5 µM 12-ADA (made 16.6 mM in DMSO) were added to the growth medium and incubated for 1–4–18 h, as indicated in the legend of the figure. The medium was removed, cells were washed three times with ice-cold water, and the cells were lysed in the flask with 1 ml of ice-cold 50 mM Tris-HCl pH 8.0, 1% SDS, and Halt™ Protease Inhibitor Cocktail (Thermo Fisher Scientific). After agitation at 4 °C for 20 min, the cell extract was collected and sonicated for 10 s on ice. The solution was then cleared of debris and aggregates by centrifugation at 16,000 g for 5 min at 4 °C. The proteins were then precipitated with methanol/chloroform and the pellet was washed with methanol, dried, and suspended in 100 µL of 50 mM Tris-HCl pH 8.0 and 1% SDS. The protein concentrations were determined with the Detergent Compatible Bradford assay kit (Thermo Fisher Scientific) with BSA as reference. About 200 µg of proteins were then reacted with alkyne-biotin, according to the manufacturer’s instructions. Proteins were precipitated with methanol/chloroform and carefully washed with methanol. Pellets were suspended in 50 µL of 50 mM Tris-HCl pH 8.0 and 1% SDS and the protein concentration was determined. Proteins (40 µg) were separated on denaturing SDS-PAGE (Any Kd TGX gel; Bio-Rad, Hercules, CA, USA), transferred to PVDF membrane and reacted with streptavidin-HRP (#STAR5B; Bio-rad). Following detection, the membrane was stripped and blotted with mouse monoclonal ACTB antibody (#sc-81178; Santa Cruz Biotech).

### 2.5. Protein Palmitoylation Quantification

Cells were grown and labeled with 10 µM 15-azido-pentadecanoic acid (Thermo Fisher Scientific) as described above. Protein extraction, biotin-alkyne reaction, and analysis were performed as described for the 12-ADA labeling experiments.

### 2.6. Fatty Acid Incorporation

Cells were grown in 96-well plates to about 70% confluence. The medium was removed and replaced by medium containing 5µM [^14^C]C_16:0_ (made as a 250 µM solution with 0.02% defatted BSA). Cells were incubated for the indicated times (10 to 120 min) and were washed twice with 0.1% BSA made in PBS to remove unincorporated labels. Fresh medium was added, and cells were incubated for one hour. For each time point, two sets of 8 wells were assayed. One set was fixed with TCA and stained with SBR for total protein quantification. Scintillation cocktail was added to the second set to dissolve the cells, which were transferred to vials and counted in a scintillation counter (LS6500 Beckman Coulter, Brea, CA, USA).

### 2.7. Acyl-CoA Synthetase and lysoPL Acyltransferase Assays

Cells grown in T75 flasks were harvested by trypsinization, washed in PBS and suspended in ice-cold 20 mM sodium phosphate pH8.0, 10 mM MgCl_2_, 5 mM DTT, 20% glycerol and Halt™ Protease Inhibitor Cocktail (Thermo Fisher Scientific). Cells were lysed with a glass grinder and debris was removed by centrifugation at 2000× *g* for 10 min at 4 °C. The protein concentration of the cleared protein extracts was determined, and the extracts were stored at −80 °C. Reactions were performed at 37 °C in 200 µL of 20 mM sodium phosphate pH 8.0, 2 mM DTT, 20 mM MgCl_2_, 10 mM ATP and 0.5 mM CoA, with 10 µM [^14^C]C_16:0_, 10 µM lysoPC (or lysoPA), and 8.5 μg to 20 μg of protein extract, as indicated in the legend of the figures. Sets of reactions were performed in the absence of lysoPC/lysoPA to assay the ACSL activity. LPCAT assays of isolated LDs were performed with 5 µM [^14^C]C_16_-CoA, 20 µM lysoPC and 0.8 μg LDs proteins. For each assay, the incubation times are indicated in the figures. Three to four time points, in triplicate, were used to determine the rates of incorporation. Calculations and statistical analysis were performed with GraphPad Prism 9.

## 3. Results

### 3.1. Role of the Acyl-CoA Synthetase Enzymes in the Myristoylation Reaction

The apparent misunderstanding of the NMT requirement for thioesterification of the acyl-donors and acyl-competitors [33] led us to reassess the role of long-chain acyl-CoA synthetase (ACSL) enzymes in the myristoylation reaction. In the cell, ACSL are responsible for the formation of the donor Myr-CoA from the fatty acid myristate, and the competitor 2-OH Myr-CoA from a non-hydrolysable fatty acid analog of myristate (2-hydroxymyristate; 2-OH Myr) [22]. As expected, we confirmed that neither fatty acid was a donor or competitor of the myristoylation reaction. Even at the high concentration of 1 mM, the formation of the myr-peptide from Myr-CoA was unaffected by the presence of 2-OH Myr (Figure 1A), and no product was formed in the presence of Myr alone (Figure 1B inset). When ATP, CoASH, and the acyl-CoA synthetase enzyme ACSL6 [44] were added to the reaction, myristoylation by NMT proceeded from Myr (Figure 1B). Under these conditions, 2-OH Myr inhibited NMT activity even at the lower concentration of 0.1 mM (≈80% inhibition Figure 1C). When myristoylation could proceed without requirement from the ACSL enzyme by providing Myr-CoA instead of the precursor Myr, 2-OH Myr could still inhibit NMT activity in the presence of ACSL (≈90% inhibition; Figure 1D). These findings confirmed the role of the ACSL enzymes in the myristoylation reaction, and the absolute requirement for thioesterification of the acyl chain. The absence of inhibition of purified NMT1 by 2-OH Myr, which was interpreted as evidence that this analog was inactive [33], was in fact expected since Myr analogs must be esterified with CoA to access the NMT binding site [17,19,20,21,45]. Synthetic compounds designed to occupy the peptide binding pocket, such as IMP-1088, strongly inhibit myristoylation, and do not require activity of cellular enzymes for their action (Figure 1A). The cellular ACSL enzymes are also responsible for the conversion of various synthetic myristate analogs, such as 12-ADA and YnMyr, into active labeling probes (see below) (Figure 2).

### 3.2. Disruption of ACBD6 in Human Cells

Disruption of the *ACBD6* gene was performed in HeLa cells using a CRISPR/Cas9 construct resulting in the out-of-frame deletion of the acyl-CoA binding domain (ACB) coding region. Several clones were selected and further analyzed for expression of an ACBD6 mRNA and protein (Figure 3). Reverse transcription of the full-length coding region in the clones produced a shorter and single cDNA (Figure 3A). Sequencing confirmed the out-of-frame deletion of exon 1 to exon 3, encoding the ACB domain (Figure 3B). The absence of an ACBD6 product in the ACBD6.KO clones was further confirmed by Western blotting (Figure 3C). Whereas disruption of ACBD6 is associated with profound neurological deficiencies in humans [3], the growth of these cells was similar to their parents, indicating that ACBD6 does not appear to be an essential protein under laboratory growth conditions.

### 3.3. NMT Activity Deficiency in the Absence of ACBD6

The availability of Myr-CoA is limiting in cells and can be out-competed by the more abundant Pal-CoA (C_16_-CoA). Formation of an ACBD6/NMT complex enhances myristoylation under acyl donor limiting conditions but also protects the Myr-CoA binding site from acyl competitors [1,2,3]. Cells not producing the ACBD6 protein provided an opportunity to assess the impact of substrate limitation and competition on the myristoylation reaction. Cells were challenged with NMT inhibitors blocking either the acyl-donor or the polypeptide binding site. The drug IMP-1088 prevents binding of the polypeptide to NMT [46,47], whereas the CoA thioester derivative of the Myr analog 2-OH Myr occupies the Myr-CoA binding site [22]. As shown in Figure 4, the absence of ACBD6 rendered the growth of the cells more sensitive to the drugs and resulted in their inability to grow even at concentrations that had little effect on the parent cells (20 nM IMP-1088 and 20 µM 2-OH Myr). The increased sensitivity of the cells to the two competitors suggested that activity of NMT was reduced in the ACBD6.KO cells and could not overcome further reduction induced by the binding of the drugs. In addition, these cells were more sensitive to the competitor targeting the Myr-CoA binding site than the peptide binding inhibitor (Figure 4A,B,D). These findings also rule out the suggestion that the growth inhibition observed in cells exposed to 2-OH Myr was the result of some non-specific toxic effect unrelated to NMT activity [33]. The absence of ACBD6 would not affect a broad metabolic defect induced by this fatty acid. In the absence of ACBD6, it appears that the NMT enzymes are no longer protected from competition and that the low abundance of Myr-CoA in the cells (≈0.1–1 µM) [48] becomes limiting when challenged with non-inhibitory acyl-donor competitor concentrations (≈10–20 µM).

### 3.4. Protein N-Myristoylation Deficiency in the Absence of ACBD6

To confirm the defect of the *N*-myristoyl-transferase reaction in the ACBD6.KO cells, the level of myristoylated proteins was quantified in vivo. Cells were grown in the presence of a Myr azide-derivative analog (12-ADA) for up to 18 h [39]. The azide-myristoylated-proteins were detected by Click chemistry with an alkyne-biotin/HRP-streptadvin system (see Section 2) (Figure 5). The transfer of the 12-ADA chain from 12-ADA-CoA onto myr-proteins resulted in a protein labeling profile similar to the pattern observed during the in vivo incorporation of ^14^C-Myr [22] (Figure 5A). Two major bands were detected within one hour labeling and additional bands accumulated over time. Compared to the parent HeLa cells, cells lacking ACBD6 had a significantly lower amount of myristoylated proteins within the first hour of labeling (≈60% of HeLa) (Figure 5B). This difference decreased during the growth of the cells and myristoylation levels of the ACBD6.KO cells were nearly as high as HeLa after 18 h (≈90%). Under laboratory growth conditions, the slower accumulation of mature myristoylated proteins from nascent polypeptides appears to be sufficient to support growth and could account for the lack of a significant growth defect of the ACBD6.KO cells (Figure 4C) [3].

To assess whether the absence of ACBD6 also slowed the activity of other protein acylation enzymes, the palmitoylation of proteins was monitored [6,10]. No significant difference in the amount and rate of protein labeling was detected in the ACBD6.KO cells compared to HeLa (Figure 5C,D). The normal level of protein acylation from Pal-CoA provided further evidence that, in the ACBD6.KO cells, the decreased transfer rate of the myristoyl chain was the result of reduced NMT activity rather than deficiency in the formation of acyl-CoAs, such as the acyl donors Myr-CoA and Pal-CoA (see below, Figure 6).

### 3.5. Role of ACBD6 in the Acyl-CoA Dependent Acylation of Lipids

Acyl-CoAs bound to ACBD6 can be channeled to the acyl-CoA dependent acyltransferase LPLAT enzymes which acylate monoacylglycerophospholipids (lysophospholipids) to phospholipids [42,49,50]. The controlled release of acyl-CoA by ACBD6 appears to be essential for the protection of these membrane-bound proteins from the detergent-like property of their substrates. The in vivo requirement of those enzymes for ACBD6 was assessed in the ACBD6.KO cells. Compared to HeLa, the levels of incorporation of exogenously added fatty acid (^14^C_16:0_) and of esterification to ^14^C_16_-CoA by the ACSL enzymes were not affected in the ACBD6.KO cells (Figure 6A,B). However, a significant defect in the acylation of lysophospholipids was observed and was not restricted to a specific LPLAT enzyme. Acylation of the two lysophospholipids LPA and LPC by the LPAAT and LPCAT enzymes was reduced by 60–70% in the absence of ACBD6 (Figure 6C,D and Figure 7).

A deficiency in the acylation of lysophospholipids, an essential step of the Kennedy and Land’s pathways, was expected to impact various processes in the cells [50,52]. The LPCAT1 and LPCAT2 enzymes are bound to the lipid monolayer surrounding the neutral lipid core of lipid droplets [53]. Synthesis of phosphatidylcholine (PC) from acyl-CoA and lysoPC by the LD-bound LPCAT enzymes is essential for LD formation and the downregulation of either LPCAT1 or LPCAT2 reduced the cellular LD content [53,54,55]. Quantification of LDs produced in the ACBD6.KO cells shows a significant depletion compared to HeLa (−30%; Figure 8A). Addition of oleic acid to the culture was successful in stimulating LD production, and the LD content of the ACBD6.KO cells increased to a near normal level (Figure 8A). The ability of the cells to respond to metabolic stimulation established that the LD synthesis pathway was not irreversibly deficient and suggested that a step had become limiting in the absence of ACBD6. Since these cells were deficient in the essential lysoPC acylation reaction (Figure 6D), lipid droplets were isolated, and their ability to acylate lysoPC from acyl-CoA was determined in vitro. The rate and yield values of PC formation of the LDs obtained from the ACBD6.KO cells were similar to those isolated from HeLa (Figure 7 and Figure 8B). The normal level of LPCAT activity bound to the LDs indicated that the decreased content in LDs of the ACBD6.KO cells was not due to diminished levels of bound-LPCAT enzymes. The absence of ACBD6 rendered the cells deficient in the Lands’ acylation pathway limiting formation of PC and LDs. These findings established a new function for ACBD6 in controlling the lipid formation of vesicles in the cells.

## 4. Discussion

The formation of an ACBD6/NMT complex enhances and protects activity of the enzyme from the binding competition of the C14 carbon donor, Myr-CoA, by abundant high binding affinity acyl-CoAs, such as C_16_-CoA or C_12_-CoA [2,14,17]. The defect in protein N-myristoylation and the increased growth sensitivity to NMT inhibitors of the ACBD6.KO cells confirmed that the absence of ACBD6 led to decreased NMT activity. Similarly, skin-derived fibroblasts of individuals carrying loss-of-function mutations of the ACBD6 gene were deficient in myristoylation and hyper-sensitive to NMT inhibitor [3]. As acyl-CoA carriers, other members of the ACBD family can weakly stimulate the NMT reaction in vitro, presumably via acyl-donor channeling, but only ANK-containing ACBD proteins can form an enzymatic complex with NMT [2]. This mechanism is not unique to human cells, and Plasmodium falciparum also has a PfACBD6/PfNMT system [2]. The finding that disruption of ACBD6 slowed but did not prevent myristoylation supports the interpretation that, as suggested, the selection process of the donor is not limited to the diffusion of the correct acyl-CoA to the donor site in the first step of the rather unique kinetic mechanism of these enzymes [11]. However, the formation of an ACBD6/NMT complex is essential to provide full activity and support the co-translational acylation modification affecting the functions and localization of an estimated thousands of proteins in human cells [11,12].

The ability of the ACBD6.KO cells to incorporate and esterify fatty acids at similar levels as compared to the parent cells confirmed that ACBD6 is not essential to the acyl-CoA transferase reactions. ACBD1 (DBI, ACBP) protects those enzymes from feedback inhibition by the acyl-CoA products and appears sufficient to support the activation of fatty acids in the cells [56,57]. These findings provided further evidence that the N-myristoylation defect observed in the ACBD6.KO cells was not a consequence of a defect in Myr-CoA formation. The decreased acylation of lipids affecting the formation of PC by the Lands’ pathway accounts for the lower production of LDs observed in the absence of ACBD6. Interestingly, downregulation of the Caenorhabditis elegans ACBD6 homolog, CeACBP-5, resulted in an opposite effect, and a 40% increase in the production of lipid droplets was observed [58]. These findings provide examples of two distinct acyl-CoA-dependent cellular processes regulated by ACBD6 either via ACB-mediated substrate binding or via ANK-mediated complex formation.

The unexpected controversy surrounding the use of compounds other than the IMP drugs designed to block the peptide-binding pocket [33] to inhibit the activity of NMT enzymes was rather puzzling. Fatty acid analogs targeting the Myr-CoA binding site of NMT must be provided as CoA thioester for binding [17,19,20,21,45]. The fatty acid 2-OH Myr is the in vivo precursor of 2-OH Myr-CoA, which is a Myr-CoA analog and a potent inhibitor of NMT (Ki of 45 nM) [22]. Esterification of 2-OH Myr with CoASH by the cellular acyl-CoA synthetases is required for inhibition of the NMT acyl-transferase reaction. In vitro, 2-OH Myr-CoA but not 2-OH Myr inhibits the activity of NMT. The Myr analog labeling probes YnMyr and 12-ADA also require thioesterification to form YnMyr-CoA and 12-ADA-CoA in order to access the Myr-CoA binding site of NMT both in vitro and in vivo [5,7,34,36,38,59,60]. Surprisingly, the inhibitory property of the activated form 2-OH Myr-CoA was not tested and the failure of 2-OH Myr to inhibit the activity of NMT1 in vitro was taken as evidence that it was not an inhibitor [33]. When added to the culture medium, 2-OH Myr has been shown to inhibit myristoylation, membrane association, and functions of diverse myr-proteins, as well as preventing their labeling with the radio-labeled [^3^H or ^14^C] Myr-CoA, which is a direct measure of in vivo N-myristoylation [3,22,23,24,25,26,27,28,29,30,31,32]. The conclusion that 2-OH Myr is not an in vitro inhibitor of NMT is accurate, but similarly, it would be accurate to dismiss the use of YnMyr and 12-ADA as myristoylation labeling probes since none of these compounds are ligands for NMT. These fatty acid analogs can cross cellular membranes and become Myr-CoA analogs once esterified by the cellular acyl-CoA synthetase. All three compounds will compete with binding of Myr-CoA to NMT. However, only the acyl chains that are not a donor of the acyl-transferase step, such as 2-hydroxymiristate, will block the catalytic cycle. The azido and alkynyl acyl chain of YnMyr-CoA and 12-ADA-CoA can be transferred on an acceptor polypeptide resulting in the labeling of myr-proteins and the release of the Myr-CoA site of the enzyme.

When assaying the in vivo inhibitory efficacy of the fatty acid 2-OH Myr and the IMP compounds on the incorporation of the YnMyr probe, cellular mechanisms interfering with those experiments should be considered. Once activated to a thioester by the cellular ACSL, the acyl chain will be targeted for incorporation into lipids during the several hour labeling period, which will limit its accumulation to levels sufficient to outcompete both the donor Myr-CoA and the probe YnMyr-CoA. The CoA thioester of YnMyr and 12-ADA bind NMT with high affinity and compete with Myr-CoA [37,38,39]. Similarly, the treatment of the cells with a high concentration of probe, 20 µM [33], will compete with 2-OH Myr-CoA for access to the Myr-CoA binding site and will protect the enzyme from inhibition. The IMP drugs target the peptide binding pocket of the enzyme [46,47] and their efficacity to inhibit NMT is not affected by the occupancy of the donor site by the YnMyr probe. Even under such unfavorable conditions, the myristoylation of a well-characterized NMT target, ARL1, was decreased in cells exposed to 2-OH Myr, but these findings were dismissed (Appendix A) [33]. As therapeutic drugs, the IMP inhibitors are likely more selective for the targeted inhibition of human intra-cellular pathogenic NMT enzymes than fatty acids analogs, but it does not change the kinetic characteristic of the reaction which will be blocked when the 2-hydroxymyristate chain occupies the donor site [19,21,22]. In addition, a ligand targeting the acyl-CoA binding site, and not the peptide binding pocket, is the appropriate NMT inhibitor to affect the steps controlling the binding and selection of the acyl-donor.

The wide range of processes affected by the members of the acyl-CoA binding proteins family, present in all kingdoms of life, cannot be solely accounted by their shared property of the binding of lipid intermediate metabolites [41]. The presence of a conserved ACB motif defined this family, but acyl-CoAs are often not involved in the affected processes, and ACBDs’ functions extend beyond trafficking and buffering of the cytosolic acyl-CoA pools. They also do not appear to have redundant functions since disruption of one is not compensated by the other members [41]. For ACBD6, the ACB domain can supply the NMT substrate Myr-CoA and sequester the competitor Pal-CoA, but in vitro, these functions are not necessary since the ANK module alone is sufficient for the stimulation and protection of NMT activity [3]. Moreover, the fusion of the ANK domain to another ACBD protein conferred the NMT-stimulatory property of ACBD6 to the chimera. These findings support the view that no essential function appears to be provided by the conserved ACB domain to the non-conserved domain. However, several findings indicate that interactions of these two domains, which can perform their functions independently, might be essential to the functions of ACBD6 in vivo. In a mixture of the Myr-CoA donor and of the Pal-CoA competitor, addition of a third preferred acyl-CoA (C_18:1_-CoA), which will saturate the ACB domain, or the phosphorylation of serine residues in the ACB domain, enhanced the activity of the ACBD6/NMT complex [1,2,3]. Although the deletion of ACB does not impair stimulation of NMT, substitution of ACB residues produced forms diminished in their ability to stimulate and protect NMT activity [1]. As a ligand of the ACB domain, acyl-CoA appears to act as a positive effector regulating the properties of the ANK domain. The ACB domain provides the acyl chain for acyl-CoA dependent processes, such as lipid acylation, but it may also act as an allosteric binding site regulating the functions of the non-conserved C-terminal motifs of the ACBD proteins. The dynamic binding properties for acyl-CoA, which is influenced by fatty acids, suggest that those characteristics are essential for ACBDs’ functions in an ever-changing environment of ligands differing in length, structure, and abundance [42]. The enhanced properties of the phosphorylated-ligand-bound ACBD6 form, which is likely the form in the cell, might be essential to overcome interference by acyl-CoAs and fatty acids on the activity of the ACBD6/NMT complex [2]. The diversity of the processes affected by ACBD6 highlight its dual function as an acyl-CoA provider and as a regulator of acyl-CoA dependent reactions controlling the lipid and protein composition of human cells membranes.

## Figures and Tables

**Figure 1 biomolecules-12-01726-f001:**
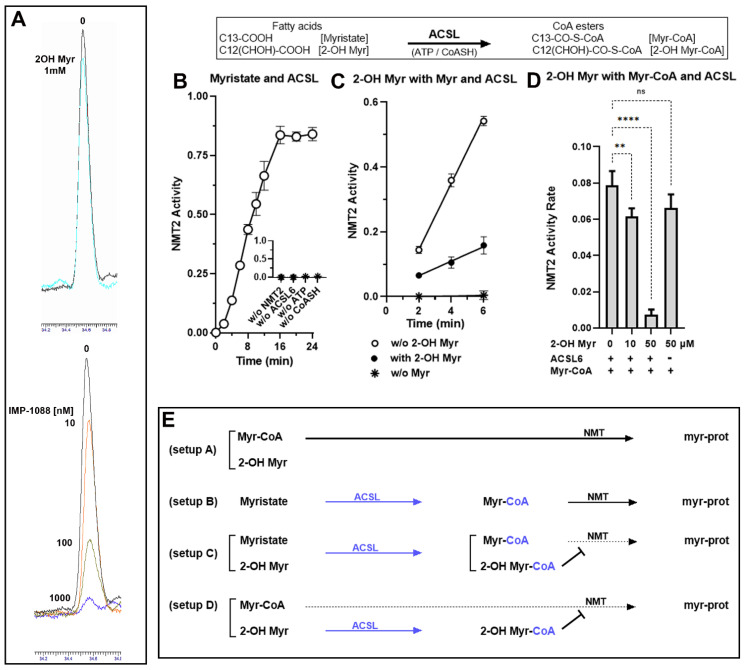
**Thioesterification requirements for acyl-donors and competitors’ formation.** (**A**) Measurements of the formation of the myristoyl-peptide were performed in the absence or presence of the drug IMP-1088 (10–100–1000 nM) or the fatty acid analog 2-OH Myr (1 mM). Reactions were performed with 250 nM NMT2 enzyme in the presence of 50 µM Myr-CoA and the indicated concentration of drugs for 20 min at 37 °C. The peptide and myristoyl-peptide were extracted and quantified by HPLC on a C18 column, as previously described [2]. The chromatogram traces obtained at 274 nm at the time of the elution of the myr-peptide (≈34.6 min) from the C18 column were overlapped, as indicated. (**B**). Measurements of the formation of the myristoyl-peptide were performed in the absence of the NMT substrate Myr-CoA but in the presence of the myristate precursor Myr and human acyl-CoA synthetase ACSL6 enzyme. Reactions were performed from 0 to 24 min with NMT2 (250 nM), Myr (20 µM) and ACLS6 (150 nM) in the presence of ATP and CoASH. Inset: Control reactions were performed in the absence of either NMT2, ACSL6, ATP or CoASH, Formation of the myr-peptide was quantified after 24 min incubation. (**C**). Measurements were performed as described in panel B but in the absence (open circle) or the presence of the fatty acid analog 2-OH Myr (100 µM) (filled circle). Control reactions were performed in the absence of the acyl-donor precursor Myr (asterisk). (**D**). Measurements were performed as described in panel C, but myristate was replaced by Myr-CoA (10 µM). The rates of myr-peptide formation were calculated and are presented as the amounts of myristoyl-peptide formed per min. Reactions were performed with the indicated concentration of 2-OH Myr in the absence or presence of ACSL6. Error bars in (**B**–**D**) represent the standard deviations of values obtained from at least three reactions: n.s., non-significant; **, *p* = 0.01; ****, *p* < 0.0001. A cartoon representation of the fatty acids and of their CoA ester derivatives produced by the action of ACSL6 in the presence of ATP and CoASH is shown in the inset above Panel (**B**–**D**). Panel (**E**). Schematic representation of the reactions presented in panels (**A**) (top traces), (**B**–**D**).

**Figure 2 biomolecules-12-01726-f002:**
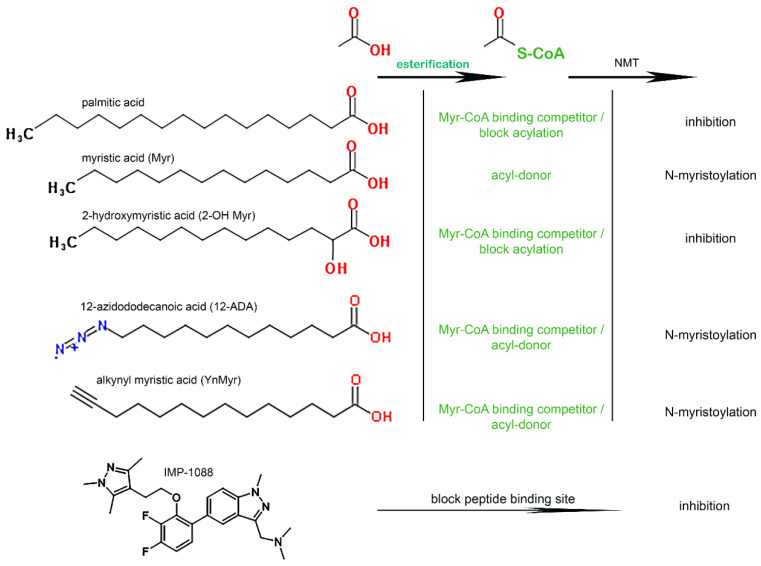
**Compounds affecting NMT activity and their precursors.** The fatty acids palmitic acid (C_16:0_), myristic acid (C_14:0_), 2-hydroxymyristic acid (2-OH C_14:0_), 12-azidododecanoic acid (12-ADA) and alkynyl myristic acid (YnMyr) are substrates of the cellular acyl-CoA synthetase enzymes. Their CoA thioester derivatives can bind the Myr-CoA binding site of NMT enzymes. The acyl-chain of C_16_-CoA and 2-OH C_14_-CoA are not a substrate for the acyl-transferase step and block myristoylation. The drug IMP-1088 (and other compounds in that class) occupies the peptide binding pocket of NMT and prevents myristoylation.

**Figure 3 biomolecules-12-01726-f003:**
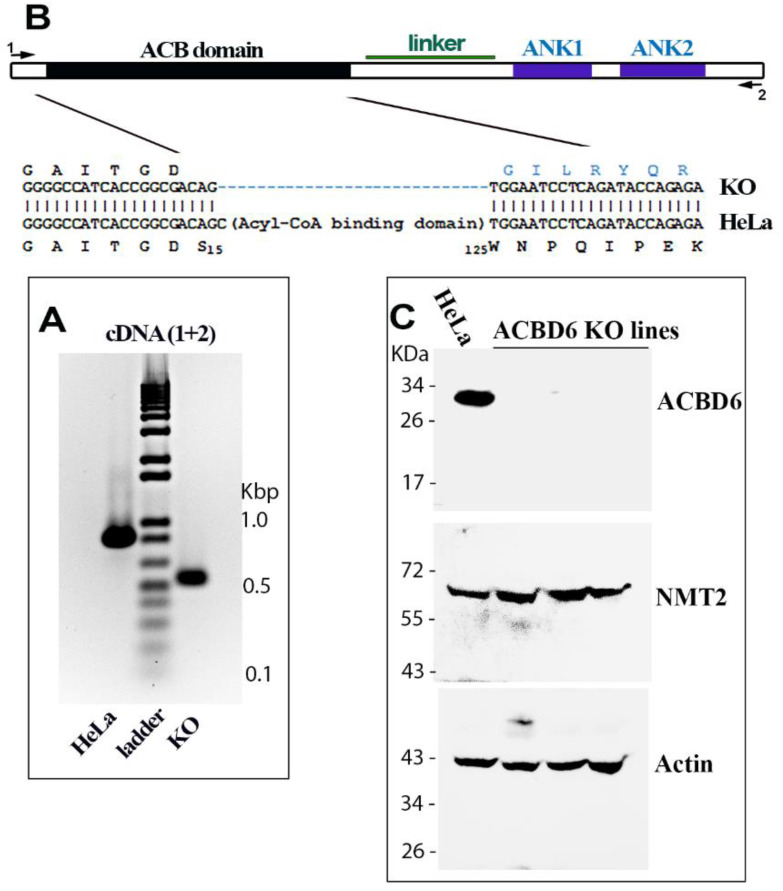
**Disruption of the *ACBD6* gene in HeLa cells.** An out-of-frame deletion of the region encoding the entire ACB domain of the *ACBD6* gene was obtained in HeLa cells. (**A**)**.** The result of PCR reactions of cDNAs obtained from RNAs isolated from the ACBD6.KO and the parent HeLa cells with *ACBD6* primers 1 and 2 is shown. The full-length *ACBD6* cDNA (≈860 base pairs) present in normal cells is absent in the ACBD6.KO cells and a single cDNA (≈530 bp) was detected establishing that no unaltered copy of the *ACBD6* gene was present in the ACBD6.KO cells. The ladder was the 1 Kb Plus DNA Ladder (ThermoFisher Scientific). (**B**). Deletion of the region coding the ACB domain was confirmed by sequencing of the cDNA. The deletion results in the out-of-frame fusion of the codons encoding Serine 15 and Tryptophan 125 preventing formation of an ACB truncated product. Primers used for the PCR amplification presented in panel A are indicated as 1 and 2. (**C**). Western-blot detection were performed with 100 µg of cell lysate proteins obtained from the parent HeLa cells and three ACBD6.KO lines. Proteins were separated on denaturing PAGE-gradient gel. After electrophoresis, proteins were transferred on PVDF membrane and probed with monoclonal antibodies against ACBD6, NMT2 and, β-actin. The protein ladder Blue Prestained Protein Standard (New England BioLabs, Ipswich, MA, USA) is indicated on the left.

**Figure 4 biomolecules-12-01726-f004:**
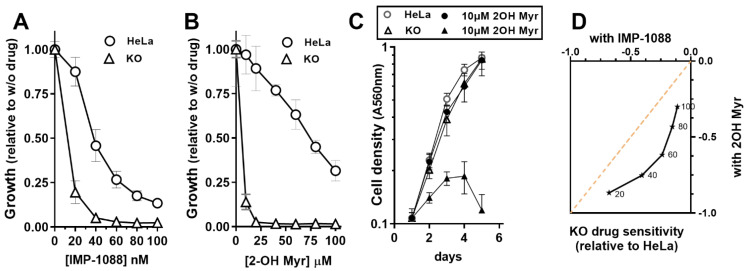
**Impaired NMT activity in ACBD6.KO cells challenged with NMT inhibitors.** HeLa and ACBD6.KO cells were grown in 96-well plates and exposed to the indicated concentration of IMP-1088 (**A**) and 2-hydroxymyristate (2-OH Myr) (**B**). Growth of cells in the absence or the presence of 10 µM 2-OH Myr are presented in (**C**)**.** Cells were fixed in 10% ice-cold TCA, stained with the SRB dye, and absorbance was read at 560 nm [3]. In (**A**,**B**), the values obtained in the presence of the drugs are reported relative to the values obtained in their absence (*p* < 0.0001). Error bars represent the standard deviations of values obtained from 12 measurements. (**D**) The values of the growth inhibition difference of the ACBD6.KO cells relative to HeLa obtained in the presence of IMP-1088 at the indicated concentrations (**A**) were plotted as a function of the values obtained in the presence of 2-OH Myr (**B**). The dashed line represents the plot of the theoretical values in a scenario of identical sensitivity for the two inhibitors.

**Figure 5 biomolecules-12-01726-f005:**
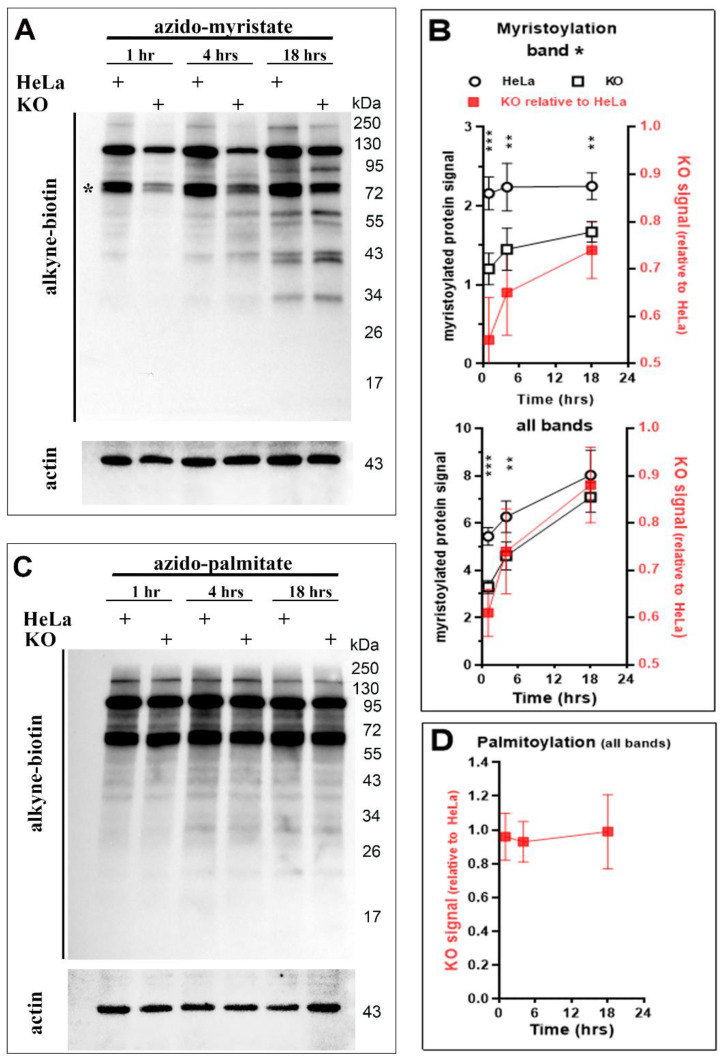
**In vivo N-myristoylation of proteins is reduced in the absence of ACBD6.** (**A**,**B**)**.** HeLa and ACBD6.KO cells were grown in flask and exposed to the labeling probe 12-ADA (azido-myristate) at a concentration of 5 µM for 1–4–18 h, as indicated. Cells were harvested, lysed, and azido-myristoylated-proteins were detected by Click chemistry (see Method). Proteins were separated on denaturing SDS-gradient gel, transferred on PVDF membrane, and detected with streptavidin-HRP. β-actin was used as a loading reference and was detected with a monoclonal antibody (**A**). Intensity of a major band (asterisk) and all the visible bands were quantified in Hela (circle) and ACBD6.KO (square) (**B**). Values are reported relative to the intensity of the β-actin signal detected in each sample as a function of time. The values obtained for the ACBD6.KO cells are also reported relative to the values obtained with HeLa cells (filled red square; data plotted on the right y axis). Error bars represent the standard deviations of values obtained from 3 measurements: **, *p* < 0.05; ***, *p* < 0.005. (**C**,**D**)**.** HeLa and ACBD6.KO cells were exposed to the labeling probe 15-azido-pentadecanoic acid (azido-palmitate) at a concentration of 10 µM for 1–4–18 h, as indicated. Proteins were detected as indicated above. The protein ladder Blue Prestained Protein Standard (New England BioLabs, Ipswich, MA, USA) is indicated on the right in (**A**,**C**). The values obtained for the ACBD6.KO cells are reported relative to the values obtained with HeLa cells. Error bars represent the standard deviations of values obtained from 3 measurements.

**Figure 6 biomolecules-12-01726-f006:**
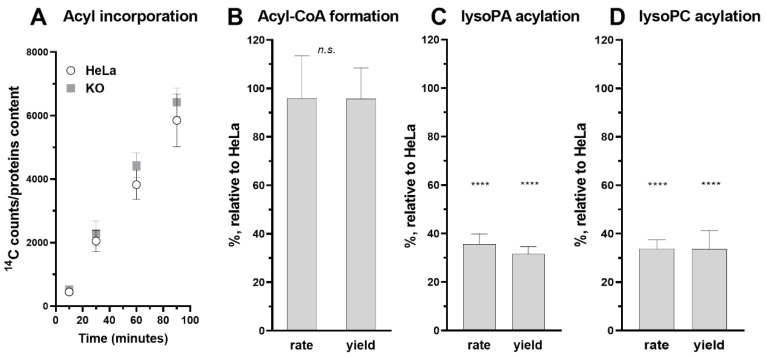
**Decrease activity of the lipid acylating enzymes.** (**A**)**.** HeLa and ACBD6.KO cells, grown in 96-well plates, were exposed to 5 µM [^14^C]C_16:0_ for the indicated times (10 to 120 min). Incorporation of the radiolabel was quantified with a scintillation counter and is reported relative to the protein content, quantified with SRB staining. Error bars represent the standard deviations of values obtained from 8 measurements. (**B**–**D**). HeLa and ACBD6.KO cells were grown in T75 flasks. Cells were harvested, lysed, and debris was removed by centrifugation. (**B**), the rate of esterification of [^14^C]C_16:0_ (10 µM) into [^14^C]C_16_-CoA in the presence of ATP, CoASH and of 8 µg protein lysate was determined from 0 to 10 min. Values obtained with ACBD6.KO are reported relative to the values obtained with HeLa. In (**C**), acylation of lysoPA (10 µM) by 20 µg protein lysate was monitored from 0 to 10 min. In (**D**), acylation of lysoPC (10 µM) by 8 µg protein lysate was monitored from 0 to 16 min. In each assay, the amount of product formed (yield) was calculated at the last time point. Error bars in (**B**–**D**) represent the standard deviations of values obtained from 6 measurements: n.s., non-significant; ****, *p* < 0.000001.

**Figure 7 biomolecules-12-01726-f007:**
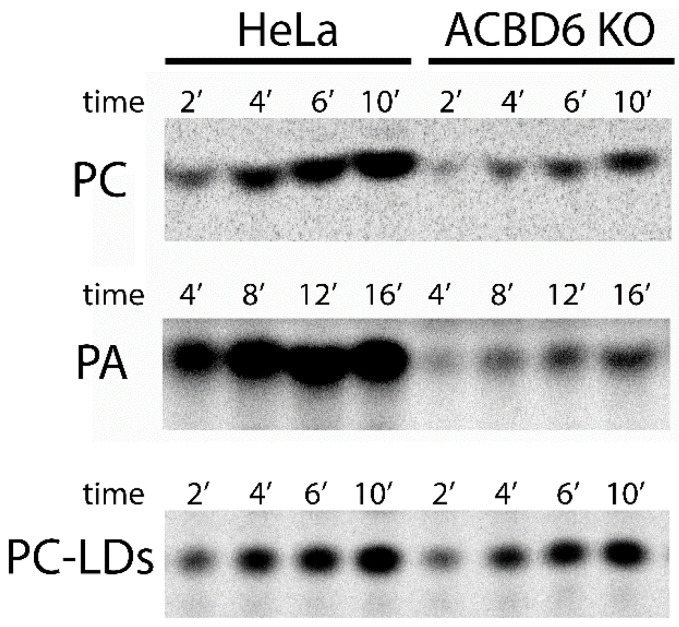
**Analysis of the acyl-CoA dependent acylation of lysoPA and lysoPC.** Products of the reactions presented in Figure 6 and Figure 8 were separated by thin-layer chromatography as previously described [51]. Formation of PC and of PA by lysates of ACBD6.KO and HeLa are shown in the top and middle panels, respectively. Formation of PC by lipid droplets (LDs) isolated from ACBD6.KO and HeLa is presented in the bottom panel.

**Figure 8 biomolecules-12-01726-f008:**
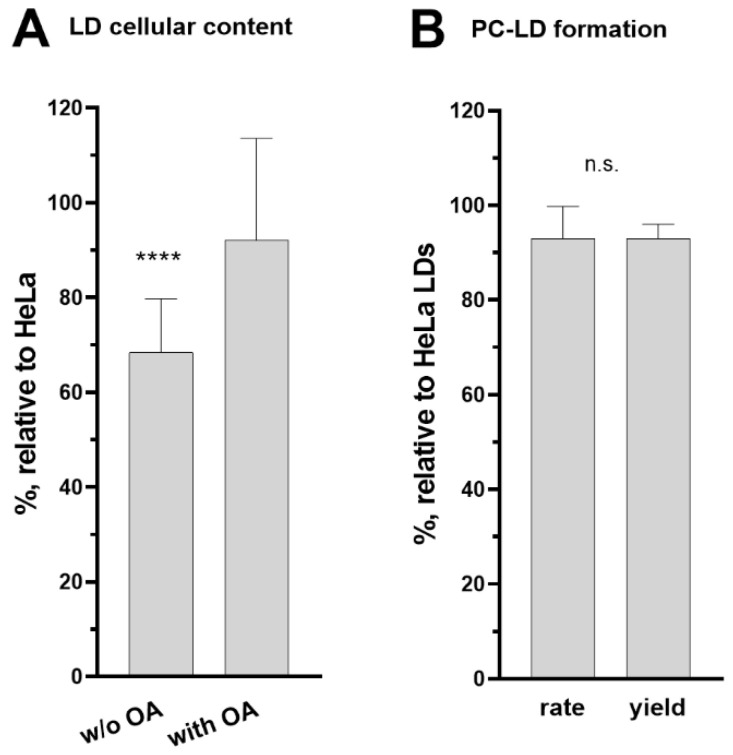
**Reduced LPCAT activity impaired LD formation in ACBD6.KO cells.** (**A**)**.** HeLa and ACBD6.KO cells were grown in 96-well plates in the absence or presence of 200 µM oleic acid (OA) for 24 h. LD content was quantified with a Fluorometric Lipid Droplet staining (see Section 2) and was normalized to the protein cellular content, quantified with SRB staining. Values obtained with the ACBD6.KO cells are reported relative to the values obtained with HeLa cells. Error bars represent the standard deviations of values obtained from 21 measurements: ****, *p* < 0.000001. (**B**)**.** Re-acylation of lysoPC (20 µM) by LDs isolated from ACBD6.KO and HeLa cells were performed with 5 µM [^14^C]C_16_-CoA and 0.8 μg LD proteins. The rate of PC formation was calculated from 0 to 10 min and values obtained with ACBD6.KO are reported relative to the values obtained with HeLa cells. Thin-layer chromatography analysis of the reactions is presented in the bottom panel of Figure 7. Error bars represent the standard deviations of values obtained from 3 measurements. n.s.—not significant.

## Data Availability

The data presented in this study are available on request from the corresponding author.

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
