# Peer review of "Dual Role of ACBD6 in the Acylation Remodeling of Lipids and Proteins"

_biomolecules, 2022, doi:10.3390/biom12121726_

Round 1

Reviewer 1 Report

Synopsis:

Acyl-CoA binding proteins catalyze lipid or protein acylation and play an essential role in lipid homeostasis and protein functions. Soupene and colleagues in this manuscript attempted to address how lipid remodeling is regulated by the reaction between acyl-CoA and acyl-CoA binding proteins, specifically ACBD6 in this study. Using the CRISPR/Cas9 gene editing technique, the authors generated an ACBD6 knockout Hela cell line, which is the novelty in this report. Using cell lysates from both WT and KO cells, the authors carried out a series of classical assays to show how lack of ACBD6 impacted NMT activity, N-myristoylation on proteins, and lipid acylation. This reviewer likes the solid data presentation from each experiment, and the results certainly provide us a better understanding of ACBD6 biology and its physiological roles. However, the general concern is that since the KO cells were able to survive like WT cells, this reviewer wonders how much of redundant functions could come from ACBD6’s homologs. If so, how confident are we to agree on the dual role conclusion as explicit for ACBD6? The following suggestions are for the authors to consider.

Suggestions:

1. How is the cell-based result compared to animal models? Phospholipid remodeling is a key point in this study; yet whole-body lipid and energy homeostasis plays an important role in phospholipid metabolism. This reviewer could not find references or discussions between these two tiers of model systems.

2. No difference was observed in the palmitoylation of the total protein preparation. Do the authors know whether the enzyme repertoires between the WT and KO cells stayed the same? If yes, any evidence; if no, could some proteomic mass spectrometry help address such gap?

3. There seems to be not much impact on lipid droplets’ formation and composition. It is known that under different metabolic conditions, the phospholipid species may change on the lipid droplets. Can the authors comment on whether ACBD6-KO cells do or do not share the same content of lipids on the lipid droplets? Say difference in lipidomic data … Any chance to observe changes in lipid droplet morphology?

4. Since the ankyrin repeat is essential for ACBD6’s role in regulating NMT, can the authors comment on what happened to ANK-less ACBD6? Same conclusion?

5. Funding: The authors declared no external funding support. Even with some internal funding mechanisms, this review believes that it is appropriate and necessary to acknowledge any support to carry out a series of experiments here.

6. Language: more rigorous proofreading is needed to correct some incomplete or mixed sentences.

Reviewer 2 Report

The manuscript by Soupene et al., studied the dual role of ACBD6 in the acylation remodeling of lipids and proteins. I will accept the manuscript in the present form with the following suggestion. 

Minor comment. 

It would be great if authors include the myristoylation process as a schematic representation in the figure 1. It will be easy for authors to follow with a schematic representation.

Author Response

We like to thank the reviewer for the suggestion. Figure 1 has been revised with the addition of a panel, panel E, presenting the reactions setup described in panels A, B, C and D. The various acyl molecules and enzymes combinations leading to the formation of the product of the reaction, myristoylated-protein, are now indicated.

Round 2

Reviewer 1 Report

The authors have addressed or clarified previous concerns.